# The Mutual Relationship among Cardiovascular Diseases and COVID-19: Focus on Micronutrients Imbalance

**DOI:** 10.3390/nu14163439

**Published:** 2022-08-21

**Authors:** Paolo Severino, Andrea D’Amato, Silvia Prosperi, Vincenzo Myftari, Aurora Labbro Francia, Merve Önkaya, Claudia Notari, Ilaria Papisca, Elena Sofia Canuti, Mia Yarden Revivo, Lucia Ilaria Birtolo, Paola Celli, Gioacchino Galardo, Viviana Maestrini, Gabriella d’Ettorre, Massimo Mancone, Francesco Fedele

**Affiliations:** 1Department of Clinical, Internal, Anesthesiology and Cardiovascular Sciences, Sapienza University of Rome, Viale del Policlinico 155, 00161 Rome, Italy; 2Anesthesiology and Intensive Care Unit, Sapienza University of Rome, Policlinico Umberto I, 00161 Rome, Italy; 3Medical Emergency Unit, Sapienza University of Rome, Policlinico Umberto I, 00185 Rome, Italy; 4Department of Public Health and Infectious Diseases, Sapienza University of Rome, 00185 Rome, Italy

**Keywords:** cardiovascular diseases, COVID-19, micronutrients, ions, vitamins, nutrition

## Abstract

Micronutrients are ions and vitamins humbly required by the human body. They play a main role in several physiological mechanisms and their imbalance is strongly associated with potentially-fatal complications. Micronutrient imbalance is associated with many cardiovascular diseases, such as arrythmias, heart failure, and ischemic heart disease. It has been also observed in coronavirus disease 2019 (COVID-19), particularly in most severe patients. The relationship between cardiovascular diseases and COVID-19 is mutual: the latter triggers cardiovascular disease onset and worsening while patients with previous cardiovascular disease may develop a more severe form of COVID-19. In addition to the well-known pathophysiological mechanisms binding COVID-19 and cardiovascular diseases together, increasing importance is being given to the impact of micronutrient alterations, often present during COVID-19 and able to affect the balance responsible for a good functioning of the cardiovascular system. In particular, hypokalemia, hypomagnesemia, hyponatremia, and hypocalcemia are strongly associated with worse outcome, while vitamin A and D deficiency are associated with thromboembolic events in COVID-19. Thus, considering how frequent the cardiovascular involvement is in patients with COVID-19, and how it majorly affects their prognosis, this manuscript provides a comprehensive review on the role of micronutrient imbalance in the interconnection between COVID-19 and cardiovascular diseases.

## 1. Introduction

Diet and a proper lifestyle are known to be very effective weapons in cardiovascular (CV) primary and secondary prevention, beyond the well-known role of pharmacological treatment [1,2]. Electrolytes, vitamins and oligoelements are involved in several physiological pathways and their imbalance has been associated with many CV diseases, such as arrythmias, ischemic heart disease and heart failure (HF). Moreover, the management of nutritional status, through the integration of micronutrients, is a main aspect to improve the general prognosis of patients, to reduce the risk of CV sequelae, and to prevent side effects caused by multidrug therapeutic approach. In this regard, the guidelines for the diagnosis and treatment of acute and chronic HF [3] suggest the importance of a precise control of micronutrient blood levels in HF patients. HF is known to be associated with sodium and potassium disorders, which relate to worse prognosis and to the unleashing of malignant arrhythmias [3].

In addition, coronavirus disease 2019 (COVID-19) is associated with electrolyte imbalance, due to the involvement of the angiotensin-converting enzyme 2 (ACE2) receptor, used by the virus to infect cells. In fact, in the most severe patients affected by COVID-19, different anomalies in fluid distribution and electrolytes concentration have been observed [4].

These subtle, but not indifferent, changes in micronutrients’ blood levels can weave a strong link between CV diseases and COVID-19. Thus, in addition to the major pathophysiological mechanisms that bind these two diseases together, increasing importance is being given to the impact of micronutrients alterations occurring during COVID-19 on the CV system (Figure 1).

Considering the frequency of CV involvement in COVID-19 patients and how it affects prognosis, the importance of studying and understanding the mechanisms that can self-amplify this relationship is substantial.

This review aims to shed light on the role of the imbalance of micronutrients in the interplay among COVID-19 and CV diseases.

## 2. Coronavirus Disease 2019 and Cardiovascular System

The most common clinical presentation of COVID-19 is a respiratory infection with interstitial pneumonia. Symptoms and signs frequently experienced include fever, dry cough, fatigue, muscle aches, shortness of breath, headache, diarrhea, indigestion, as well as fluid and electrolyte imbalances [4].

Cardiovascular diseases have high prevalence in patients with COVID-19. In particular, arterial hypertension can be found in 56.6% of patients, diabetes mellitus in 33.8%, and acute myocardial injury in 12% of hospitalized patients [5,6]. Patients with CV diseases are prone to develop a more severe form of COVID-19 [5,6,7]. The main CV symptoms experienced by COVID-19 patients are heart palpitations and chest tightness/pain, mostly experienced as early manifestations of the infection [8], but also during long COVID-19 [9,10,11,12]. There is strong evidence to support the interconnection among CV pathologies and COVID-19, leading to high rates of CV morbidity and mortality. In this regard, different hypotheses have been advanced: (a) direct CV injury determined by the virus, (b) aggressive immune response leading to a cytokine storm, (c) hypoxic injury due to severe pneumonia, (d) psychological injury (depression, fear, stress, anxiety) with noradrenergic overactivation, (e) vascular thrombosis, and (f) cardiac damage induced by drugs administered for COVID-19 [7,8,13,14]. Data show that COVID-19 patients, especially those ones affected by pre-existent CV diseases, are prone to develop CV complications, including myocardial injury, myocarditis, myocardial infarction, cardiac arrhythmias, cardiac arrest, venous thromboembolic disease, HF, and Takotsubo cardiomyopathy [15,16,17,18,19,20].

The mechanism of acute myocardial damage is half immune and half ischemic. They are connected to each other. Heart biopsy of COVID-19 patients showed degeneration of myocardial cells, necrosis and presence of inflammatory infiltrates [8,21,22,23,24], showing that myocardial injury is related to the inflammatory reaction against the virus. However, inflammation can also predispose to coronary atherosclerotic plaque rupture. Acute myocardial infarction was reported to be one of the common cardiac complications in COVID-19, together with arrythmias [8]. For instance, cardiac arrhythmias may result from hypokalemia, which is in turn stimulated by the known interaction between COVID-19 and the renin-angiotensin-aldosterone system (RAAS) [25,26].

Moreover, venous thromboembolic disease is promoted by the inflammatory state, but at the same time it predisposes to a hypercoagulability state [8,27,28,29].

As regards the relationship between HF and COVID-19, some hypotheses include deterioration of a pre-existing cardiac dysfunction, but also a new-onset cardiomyopathy or myocarditis, pulmonary hypertension, induced by lung involvement, infection, and acute pulmonary embolism [8,24,30].

Another major aspect to be concerned about when considering the effects of COVID-19 on the CV system is connected to fever. We know that fever is often part of the clinical presentation of symptomatic COVID-19 infections. Fever is known to relevantly affect the CV system first by leading to hypotension, which is most likely due to a redistribution of blood and to nitric oxide (NO)-induced vasodilation. Fever may also lead to several electrocardiogram abnormalities, including conduction defects, ST and QT changes, T wave abnormalities, and malignant arrhythmias. Moreover, it has also been associated to cardiac dysfunction and pulmonary oedema. Additionally, inflammatory response affects myocardium inducing cardiomyocytes damage [31]. Hyperthermia is also known to cause a plethora of metabolic abnormalities, including hypoxia, respiratory alkalosis, metabolic acidosis, and hypoglycemia. It can massively alter electrolytes homeostasis, leading to wide electrolytes imbalance [32]. All these pathophysiological alterations are shared among cardiovascular disease and COVID-19. In fact, the frequent myocardial involvement occurring in COVID-19 is demonstrated by the increase in relevant cardiac biomarkers, including creatine kinase isoenzyme-MB (CK-MB), N-terminal pro-brain natriuretic peptide (NT-proBNP), and high-sensitivity Troponin I (hs-cTnI) and T (hs-cTnT) [24,33].

COVID-19 has definitely impacted on CV diseases. It has represented a challenge for cardiologists worldwide and has forced the adoption of the maximal deployment of health resources, not only in the acute hospital scenario, but also in the outpatient setting [34,35,36,37].

## 3. Coronavirus Disease 2019 and Cardiovascular Diseases: The Role of Ions

Ions play a pivotal role in the maintenance of homeostasis in the human body, carefully regulated by their distribution in the intracellular and extracellular fluids. Sodium is the main electrolyte in the extracellular space (normal serum range: 135–145 mmol/l). On the other hand, potassium is mainly an intracellular ion (normal serum range: 3.6–5.5 mmol/l). Calcium is mostly present in the extracellular fluid (normal serum range: 8.8–10.7 mg/dL), whereas magnesium is mostly found in the intracellular compartment (normal serum range: 1.46–2.68 mg/dL). As for iron, despite being an essential trace element, it may be highly toxic in excess amounts, for which reason there are several mechanisms aimed at keeping its cellular and whole-body concentrations at the optimal range (body iron amounts range: 3–5 g). For what relates to zinc (normal serum range: 12–16 µM), it is mainly bound to albumin, α-macroglobulin, and transferrin, with only a sub-nanomolar concentration of free. Derangements in ion levels are associated with CV diseases, spanning from arrhythmias to atherosclerosis burden, from arterial blood pressure alterations to coagulation cascade imbalance. Interestingly, electrolyte disturbances have been reported in a substantial number of COVID-19 patients, mainly in the most severe patients, showing a significant impact on their prognosis. 

### 3.1. Sodium

Sodium (Na+) is an essential cation that plays a major role in the regulation of blood volume, blood pressure, osmotic and acid-base equilibrium. Dysnatremia is the most common electrolyte disorder, with a prevalence between 15% and 20% in the hospital setting [38]. Furthermore, it is an independent risk factor for mortality, in hospitalized patients [39]. These data are also confirmed in COVID-19 patients, in which low Na+ levels are associated with an increased risk of encephalopathy and treatment with mechanical ventilation [40]. The most common cause of hyponatremia is the inappropriate antidiuretic hormone secretion syndrome (SIADH), which accounts for up to 40–50% of cases [41], particularly in patients with atypical viral pneumonia, such as COVID-19 [42]. Interestingly, desmopressin, the synthetic form of the antidiuretic hormone, also has a role in the release of coagulation factor VIII and von Willebrand factor by platelets. For this reason, it is used as a procoagulant treatment, for example, in von Willebrand’s disease. Therefore, the increased release of antidiuretic hormone, also called arginine vasopressin, in patients with SIADH-related hyponatremia may be one of the causes leading to the procoagulant state observed in COVID-19 [43].

Moreover, it has been proven that pro-inflammatory cytokines, such as Interleukin 1β (IL-1β) and 6 (IL-6), known to be the key components of the cytokine storm associated with COVID-19, can stimulate hypothalamic vasopressin secretion [44], leading to hyponatremia. This seems to suggest that low levels of Na+ may be associated with a stronger inflammatory condition. In fact, in a small retrospective study, it was shown that IL-6 levels were inversely proportional to serum Na+, with the lowest natremia in patients exhibiting the highest IL-6 levels [45].

However, even though hypernatremia is less common [46], this condition is a negative prognostic marker in critically-ill patients in intensive care unit (ICU) [47]. It has been shown that especially hospital-acquired hypernatremia is a predictor of mortality [48]. Hypernatremia reflects a deficit in total body fluids and, therefore, it suggests that volume depletion is a possible pathophysiological mechanism associated with poor outcomes.

The issue of fluid balance is crucial both in patients with COVID-19, who often present dehydration, and in HF, where patients frequently have fluid excess in the third space and are excessively dehydrated from the use of diuretics. Even if rehydration and volume repletion could be a reasonable therapeutic approach, conservative fluid regimens often need to be applied as a component of lung-protective strategies. This is also true in HF patients, where intravenous rehydration therapy must be administered more cautiously to avoid patient’s fluid overload.

In conclusion, natremia appears to deserve a strict control in both COVID-19 and CV patients, especially in those undergoing ion-depleting treatments, such as proton-pump inhibitors, diuretics, angiotensin-converting enzyme inhibitors (ACEi), angiotensin II receptor blockers (ARBs), and non-steroidal anti-inflammatory drugs [49].

### 3.2. Magnesium

Magnesium (Mg++) is the fourth most abundant electrolyte in human cells, and it is mainly concentrated in the mitochondria. It is essential for basic biochemical reactions, participating in a cluster of physiological functions, such as cell cycle, muscle contraction, vasomotor tone [50], energy metabolism, and protein and nucleic acid synthesis [51]. Mg++ has anti-inflammation [52] and antioxidant potential [53]. It may induce vasodilation [54], neuroprotection [55] and immunomodulation [56].

Mg++ importance in CV physiology is well-known. It has been demonstrated that its deficiency is associated with an increased incidence of CV diseases [57], in particular arrhythmias, hypertension and atherosclerosis [58]. In fact, its supplementation can lower blood pressure [59], due to Mg++ having a similar property as the one of calcium-channel blockers, making it a physiological calcium antagonist [60]. It also participates in activation of K+ channels, promoting the cellular outflow of calcium (Ca++) ions and it inhibits cellular Ca++ influx and release from the sarcoplasmic reticulum. The result is membrane hyperpolarization and vasodilation. Moreover, it promotes increase in serum nitric oxide (NO) levels, playing an important role in physiological endothelial function and vascular smooth muscle cells relaxation [61,62]. Alterations of Mg++ blood levels as hypomagnesaemia have been found in patients hospitalized with COVID-19, appearing to be significantly correlated to patient’s prognosis. In fact, hypomagnesaemia has been associated with more severe cases of COVID-19 [63]. Alamdari et al. [64] demonstrated that patients with hypomagnesemia at hospital admission were at higher risk of mortality due to the COVID-19. These findings may be related to the high incidence of CV complications in COVID-19 patients, in which serum Mg++ levels may play an important stabilizing role. Furthermore, it has been demonstrated that Mg++ deficiency promotes endothelial dysfunction, increasing the susceptibility of endothelial cells to oxidative damage [65,66]. Endothelial dysfunction has a main role in coronary microvascular dysfunction and HF with preserved ejection fraction (HFpEF) [15,28,67,68,69,70,71].

Mg++ also has antithrombotic effects. In this regard, hypomagnesemia is associated with increased thrombotic risk and slowed fibrinolysis [72,73]. It reduces platelet aggregation, and it increases blood-clotting times in vivo [74]. Moreover, with its important role in membrane stabilization, Mg++ may help to prevent and treat arrhythmias, commonly observed in COVID-19 patients, and associated with negative prognosis [75].

Mg++ has also indirect benefits on the CV system, through its effect on the respiratory system. As a Ca++ antagonist, it inhibits bronchial smooth muscle contraction, promoting bronchodilation [76,77]. Moreover, through its anti-inflammatory and antioxidant power, it contrasts lung inflammation by inhibiting cytokines, such as IL-6, nuclear factor kappa-light-chain-enhancer of activated B cells (NF-κB) pathway, and Ca++ channels [78].

### 3.3. Potassium

Potassium (K+) is the most represented cation in the human body, with a 98% intracellular location, and only the remaining 2% in the extracellular space [79]. The intracellular K+ is critical for cell volume regulation, protein synthesis and transmembrane gradient creation. K+ is also a key determinant of the membrane potential, being essential for ion transportation across the membrane. Furthermore, it is involved in several physiological mechanisms [80], such as vascular tone and systemic blood pressure regulation, gastrointestinal motility, acid–base homeostasis, glucose, and insulin metabolism [81,82]. Several intra- and extra-renal mechanisms are involved in maintaining the K+ serum concentration within a narrow physiological range between 3.5 and 5.0 mEq/L [83].

K+ balance disorders are common among hospitalized patients, with an incidence of hypokalemia up to 21% compared to 2–3% of outpatients. Less common is hyperkalemia, which is detected in up to 3.3% of hospitalized patients, compared to 1% of outpatients [84,85]. Several studies have found a high incidence of K+ imbalance in COVID-19 patients, which seems to be one of the most common electrolyte disorders [86,87,88], independently affecting disease prognosis [89,90]. Tezcan et al. [91] reported high prevalence of hypokalemia among COVID-19 patients. Chen et al. [89] reported that hypokalemia was associated with disease severity. Furthermore, significantly lower levels of serum K+ in confirmed cases of COVID-19 compared with non-infected patients have been reported [92].

A plausible mechanism linking COVID-19 to hypokalemia could be related to ACE2 receptor degradation, after the virus has entered in host cells [93]. The consequent downregulation of ACE2 leads to an increased activity of RAAS, with hypokalemia related to potassium wasting through urine [94]. In addition, viral-mediated neurodegeneration and neuroinflammation may cause hypothalamic paraventricular nucleus and supraoptic nucleus dysfunction and reduced antidiuretic hormone production, leading to urinary potassium loss and hypokalemia [95]. Hypokalemic patients affected by COVID-19 have longer hospitalization and ICU permanence. Moreover, hypokalemia is prevalent in patients with COVID-19 pneumonia, and it represents an independent predictor of invasive mechanical ventilation [96]. Liu et al. found out that severe and moderate hypokalemia were associated with ICU admission [90].

Regarding hyperkalemia, it seems to be more related to kidney injury, which is not rare in hospitalized COVID-19 patients [97]. Huang et al. [98] found higher K+ levels in ICU patients than in patients not admitted to the ICU, showing that also elevated serum K+ is associated with the severity of illness. All these observations may be related to potentially life-threatening complications, such as cardiac dysrhythmias, paralysis, and rhabdomyolysis [99,100].

Coromilas et al. [101] conducted a retrospective analysis, collecting data of COVID-19 patients, and found out that almost 18% had new-onset arrhythmia, 43% of patients who developed arrhythmias were mechanically ventilated, and only 51% survived to hospital discharge. Elias et al. [102] reported arrhythmias’ prognostic value as regards early deterioration in patients with COVID-19 and the association with significant morbidity and mortality.

Hence, the investigation of the modifications of K+ serum level, its consequences on COVID-19 patients and the continuous CV monitoring in those patients is crucial [103].

### 3.4. Calcium

Ca++ plays an important role in several systemic functions. It’s involved in coagulation and platelet adhesion, myocardial contractility, and relaxation. Proteins involved in the coagulation system, such as C and S protein, as well as clotting factors II, VII, IX, and X, need calcium for their activation. Several studies demonstrated that there was a very high prevalence of hypocalcemia in patients affected by COVID-19 with a negative prognostic value [104,105].

Di Filippo et al. [106] investigated if hypocalcemia was a COVID-19 specific parameter, finding out that COVID-19 patients had lower calcium levels compared to patients without COVID-19, with a doubled rate of hypocalcemia. Other studies showed a lower Ca++ level in COVID-19 patients compared to the negative control group [107,108].

The mechanism through which COVID-19 is associated with serum Ca++ imbalance is still unknown. A potential factor may be gastrointestinal loss due to diarrhea and vomit [109]. Malnutrition and hypovitaminosis-D could reduce the absorption of calcium too. It can be also hypothesized that high viral load may cause Ca++ depletion because calcium ions are involved in viral life processes, such as regulating virus cell-entry, gene expression, and virion formation [110].

Ca++ plays an important role in maintaining heart and coagulation function [111,112,113]. Dysfunction of coagulation/fibrinolysis system can cause diffuse intravascular coagulation, which is a decisive factor in the death of COVID-19 patients and often related to the inflammatory cascade in which Ca++ levels can be involved [114,115,116].

It has been demonstrated that the level of coagulation factors like D-dimer and prothrombin is higher in hypocalcemic compared to normocalcemic COVID-19 patients and that there is a negative correlation between serum calcium and D-dimer levels [117,118].

Qi et al. [119] compared two groups of COVID-19 patients, one made up of patients with mild manifestations and the other made up of severe cases. They demonstrated that severe patients with a worse inflammation status related to a higher level of procalcitonin showed lower levels of Ca++. The severe cases showed higher levels of D-dimer and fibrin degradation product that could be related to a more severe coagulation dysfunction and consequently to a higher risk for thrombotic complications. Furthermore, severe patients showed a worse inflammation status related to a higher level of procalcitonin and calcitonin, which may explain why the Ca++ serum concentration was significantly reduced. In conclusion, decreased Ca++ levels and coagulation dysfunction in COVID-19 patients were related to each other and with the inflammatory status [119].

Due to its relationship with the inflammatory response and severity of COVID-19 manifestations, hypocalcemia has been related to a higher mortality. Several study cohorts demonstrated the relation between lower Ca++ levels at hospital admission and higher risk of in-hospital and 28-day mortality [91,120,121]. Limiting severe acute hypocalcemia could protect from CV and neurological complications that may be fatal. For this reason, a good supplementation of calcium and albumin, but also of Vitamin D, may reduce multiorgan injury [122].

### 3.5. Iron

Iron is a key element in major human biochemical processes, such as deoxyribonucleic acid (DNA) replication, cellular respiration, and immune defense. Iron can donate or accept electrons because it can shift in two different oxidative forms: divalent (Fe++) or trivalent (Fe+++). However, its blood level needs to be tightly controlled as a low iron level as well as a high blood level can be very dangerous [123].

Iron is generally not an easily-accessible element in the human body. It is mostly stored safely inside the erythrocytes. A low percentage of extracellular iron travels bound to transferrin or to other molecules. Iron is a growth factor and an energetic source, also for pathogens. Many microbes use hemolysis as a damage mechanism to afflict the host and to recover iron sources. The consequent massive release of free iron into the circulation can also cause oxidative damage and immunity inhibition [124,125]. For self-protection during inflammation, macrophages and neutrophils are induced by cytokines to produce hepcidin, responsible for circulating iron levels reduction [126,127]. This is in line with many COVID-19 studies [122,123,124]. In fact, patients admitted to hospital for COVID-19 had a lower level of hemoglobin than other patients [128,129].

Fan et al. [130] compared laboratory data between patients with COVID-19 admitted to ICU and not, finding a significantly lower hemoglobin value in the first group.

In the meta-analysis of Taneri et al. [128], a significant lower hemoglobin value was found in more severe COVID-19 group. Moreover, this difference was greater in patients with systemic arterial hypertension or severe form of infection. In fact, during chronic and prolonged infections, a vicious circle is thus created: the organism tries to defend itself by decreasing the iron levels, but the resulting anemia can aggravate the infection itself, as well as its comorbidities and complications. For this reason, iron therapy during infectious diseases is much debated. In this context, the reduction of iron intake by uptaking chelating agents as adjuvant COVID-19 therapy has been suggested, considering that severe acute respiratory syndrome coronavirus 2 (SARS-CoV2), like other viruses, requires iron to carry out replicative cycles [131,132]. In fact, hereditary diseases with iron storage dramatically increase susceptibility to severe infections. In this regard, several studies have shown that iron deficiency is crucial in depressing the immune response [131,133], even more impactful in special populations. In children, for example, anemia is significantly associated with repeated infections of the respiratory and gastroenteric tract [131,134].

Furthermore, HF patients may benefit from iron supplementation [3,131,135]. Iron deficiency and anemia are frequent comorbidities in CV diseases. In HF they represent an independent condition predisposing to CV and overall mortality, rehospitalization, quality of life and symptoms worsening [3,136,137,138]. In HF patients, iron deficiency affects about 50% of the population with a prevalence of 80%, in patients with acute HF [139,140,141,142,143]. Several mechanisms are associated with iron deficiency in both heart failure with reduced ejection fraction (HFrEF) and HFpEF, such as reduced absorption, loss and hyperactivation of systemic inflammatory response [144,145]. The importance of iron supplementation with ferric carboxymaltose in HF patients has been demonstrated [146,147]. Beyond HF, the impact of iron deficiency and the effects of iron supplementation have been evaluated in other CV diseases, which have been often described as COVID-19-related CV complications. As regards, the Effect of Iron Repletion in Atrial Fibrillation study (IRON-AF) will evaluate the role of ferric carboximaltose supplementation in patients with atrial fibrillation and iron deficiency [148]. Iron deficiency has been also described by Vinke et al. in patients with pulmonary arterial hypertension and chronic thromboembolic pulmonary hypertension [149]. Moreover, in the setting of atherosclerosis and coronary syndromes, iron may have a pivotal role. It is involved in oxidative stress, low-density lipoprotein (LDL) oxidation and inflammation [150]. Meng et al. [151] reported that reduced circulating iron values were a predictor of coronary atherosclerosis.

### 3.6. Zinc

Zinc (Zn++) is a vital mineral involved in many physiological processes in the human body. First, it contributes to cellular immunity against bacteria and viruses by activating NF-kB, regulating inflammatory cytokine release [152]. Second, it plays a role in the synthesis of NO and in the suppression of reactive oxygen species (ROS), during inflammatory processes [153]. As a result, Zn++ deficiency is associated with oxidative stress [154]. NO signaling promotes vasodilation, increased blood flow, and a reduction in plaque development and progression [155,156]. When this pathway is disrupted, together with abnormal ROS and NF-kB signaling, the groundwork for atherosclerosis is laid [157]. Thus, it is hypothesized that Zn++ may minimize atherosclerotic disease by increasing NO production and NF-kB signaling, as well as by reducing oxidative stress induced by endothelial dysfunction [158].

Zn++ has also been demonstrated to exert an inotropic impact on the heart, inhibiting cardiomyocyte systolic activity and increasing relaxation function by lowering intracellular Ca++ levels [159].

Due to Zn++ related antiviral and pro-cardiogenic qualities, it may have a pivotal role in COVID-19 patients, particularly those with CV complications [158]. A critical factor supporting this hypothesis is the fact that the SARS-CoV-2 receptor, ACE2, is a Zn++-regulated metalloprotein [158]. ACE2 works by converting angiotensin II to angiotensin (1–7), causing vasodilation [158]. ACE2 is an important player in the RAAS because it inhibits the actions of Angiotensin II [160]. The lowering ACE2 expression, often observed in COVID-19 patients, has a negative impact on a variety of diseases, such as arterial hypertension, diabetes mellitus, and CV diseases [161,162]. Extra Zn++ supplementation may help to activate or upregulate functional ACE2 expression, restoring the functional balance of ACE2 [158].

Another mechanism by which Zn++ may be beneficial is by inhibiting SARS-CoV-2 replication through the inhibition of elongation and template binding of SARS-CoV RNA-dependent RNA polymerase [163]. Zn++ is also essential for autophagy, which is especially important during viral infections [158]. According to a prospective research, Zn++ deficiency was associated with more serious complications and an increased length of hospital stay and death, in COVID-19 patients [164].

A summary of ion imbalance effects on CV diseases and COVID-19 is represented in Table 1.

## 4. Coronavirus Disease 2019 and Cardiovascular Diseases: The Role of Vitamins

Both lipo-soluble and hydro-soluble vitamins are intimately involved in the regulation of different metabolic pathways, concerning energy production, cellular homeo-stasis and clearing of catabolites. Consequently, deficiency in some vitamins is associated with CV diseases, whereas supplementation in these vitamins has been proven beneficial in preventing different kinds of CV alterations. The same is true for COVID-19, where it has been seen that lower levels of some vitamins were linked to worse outcomes.

### 4.1. Vitamin A

Vitamin A is a fat-soluble vitamin with a broad range of immunological effects. Its deficiency is associated with recurrent infections [165]. The protective role of vitamin A supplementation against various infections has been demonstrated [165]. In this regard, significantly lower vitamin A serum levels in patients with severe COVID-19 symptoms have been observed [166]. Most likely, this is due to the inflammation itself rather than a direct effect of vitamin A deficiency during the illness, suggesting that supplementation may be needed to restore the normal status [167]. A positive association between plasma retinol and dyslipidemia has been pointed out [168], as well as a significant reduction of overall CV and respiratory-disease-related mortality in male patients with higher levels of vitamin A [169].

### 4.2. Vitamin B

The vitamin B complex is made by eight different water-soluble constituents that humans must absorb from the diet. Deficiency of B12 and B9 is mostly known as it results in megaloblastic anemia. However, a significant role of vitamin B, particularly B6, B9 and B12, in the immune response has been demonstrated too [170]. In particular, it has been shown that vitamin B12 could suppress viral replication in the host cells [171]. Dos Santos et al. hypothesized a therapeutic role of vitamin B12 because it could reduce severe CV sequelae of COVID-19, through the reduction of oxidative stress, anti-inflammatory and analgesic effect [172]. It is well known that vitamin B6, B9 and B12 also contribute to the reduction of blood homocysteine concentrations, which are related to an increased risk of coronary artery disease and stroke. [173,174]. Similarly, deficiency in B1 has been associated with a higher prevalence of CV risk factors as dyslipidemia, obesity and diabetes [175]. Interestingly, B9 supplementation has been shown to improve the endothelial dysfunction by increasing endothelial nitric oxide production and by scavenging superoxide radicals [176], which suggests a possible beneficial role in COVID-19 patients, where endothelial dysfunction is known to be one of the leading causes of mortality. However, despite this theoretical evidence, vitamin B group supplementation has no significant effect on major adverse CV events, overall mortality, cardiac death, myocardial infarction, and stroke [177], while B12 supplementation was shown to decrease the severity of COVID-19 symptoms [178].

### 4.3. Vitamin C

Vitamin C is known for its antioxidant effects and its role in immunomodulation. Due to its ability to readily donate electrons, it protects LDL from oxidation. Different studies have shown its role in reducing harmful oxidants in the stomach and promoting iron absorption [179], as well as protecting cells during exposure to toxins and pollutants [180].

As for its immunomodulation effect, its action on lymphocytes is not completely clear, but it has been proven that vitamin C regulates genes responsible for B and T cell generation, differentiation, and proliferation. Moreover, amongst its pleiotropic effects, it has been reported to have an antithrombotic function by inhibiting platelet expression of CD40 ligand [181,182] and another important role as co-factor to produce catecholamines, vasopressin, norepinephrine, and cortisol, in the human body [183,184]. This last evidence provides the rationale for evaluating intravenous vitamin C administration in septic shock to achieve a sparing effect on vasopressor requirements [177]. Early trials have indicated some potentially beneficial effects of intravenous vitamin C also in severe COVID-19 patients [185,186]. The upcoming findings from the larger randomized control trials currently underway, such as the Lessening Organ Dysfunction with Vitamin C-COVID (LOVIT-COVID) trial, should provide more definitive evidence (NCT04401150).

### 4.4. Vitamin D

Vitamin D plays a central role in the absorption of Ca++, Mg ++, and phosphate, as well as in the bone metabolism. Strong evidence suggests a critical role for vitamin D in the modulation of the immune function [187]. Furthermore, vitamin D deficiency has been reported in several chronic conditions associated with increased inflammation and deregulation of the immune system, such as diabetes mellitus, asthma, and rheumatoid arthritis [187]. Similarly, an association between vitamin D deficiency and inflammation was shown in patients with COVID-19. In a prospective study by Jain et al. [188], serum 25(OH)D concentrations were significantly lower in patients requiring ICU admission than in asymptomatic patients with COVID-19. Moreover, epidemiological data indicate that vitamin D deficiency in humans is associated with arterial stiffness, arterial hypertension, left ventricular hypertrophy, and endothelial dysfunction, implying that vitamin D might also have a protective role in CV diseases, and that it may lower the risk for HF [189].

It is well known that diabetes mellitus is associated with an increased risk for severe COVID-19, as hyperglycemia might modulate immune and inflammatory response leading to possible lethal outcomes [190]. Vitamin D insufficiency plays a negative role in this dysregulation, leading to impaired glucose homeostasis. In fact, Vitamin D is essential for pancreatic β-cell functioning and insulin sensitivity. It plays a role in controlling gene transcription and cell signaling, alleviating the onset of insulin resistance, especially in adipose tissue [191].

For what concerns the CV system, data from several experimental studies support the anti-fibrotic and anti-hypertrophic role of Vitamin D, suggesting that it has a beneficial role against cardiac dysfunction, hypertrophy, and fibrosis [192,193].

Vitamin D is also involved in the regulation of the RAAS even though the mechanism is not completely understood [194]. It has been reported that vitamin D3 supplementation reduces blood pressure in patients with essential arterial hypertension [195,196] and 1,25(OH)_2_D_3_ treatment reduces blood pressure, plasma renin activity, and angiotensin II levels, in patients affected by hyperparathyroidism [197,198]. Moreover, it has been shown that Vitamin D is also involved in the regulation of thrombotic pathways, and its deficiency is associated with an increase of thrombotic episodes probably because it allows the production of certain neuroprotective growth factors and inhibition of ROS [199,200]. This finding is of crucial importance when looking at COVID-19, since vascular thrombosis is one of the most serious complications often leading to death.

### 4.5. Vitamin E

Vitamin E is a fat-soluble vitamin, primarily known for its role as an antioxidant. It appears to prevent the polyunsaturated fatty acids (PUFAs) in the cell-membrane from oxidation, regulate the production of ROS and reactive nitrogen species, and modulate cell signaling. Immune cells are particularly sensitive to oxidative damage due to their high metabolic activity and PUFAs contents [201]. Therefore, vitamin E plays an important role in fostering the immune system. Vitamin E has also been shown to have suppressor effects on prostaglandin E2 (PGE2) synthesis, which is an important T cell-suppressing mediator.

Furthermore, it is well known that one of the key complications of the respiratory distress syndrome associated with severe COVID-19 is acute cardiac injury [202], which is associated with oxidative stress, characterized by enhanced nicotinamide adenine dinucleotide phosphate oxidase2 (NOX2) activity, activation of platelets phospholipase A, and excess of PGE2 [203].

Moreover, other effects of Vitamin E on CV diseases have been investigated in basic science trials, showing a preventive role of α-tocopherol, the biologically most active form of vitamin E, in CV diseases [204,205,206]. These beneficial effects include inhibition of cell proliferation and LDL oxidation by modulating protein kinase C (PKC) activity [204], protection against atherosclerotic lesion development and aorta damage [205], and a reduction of apoptotic activity in cardiomyocytes [206].

Thus, because of its proven antioxidant effects and its theoretical beneficial effects against plaque formation, an anti-oxidative therapy with Vitamin E could be proposed to reduce the burden of cardiac complications caused by COVID-19 [207].

### 4.6. Vitamin K

Vitamin K is a fat-soluble vitamin necessary for the synthesis and activation of both procoagulant or anticoagulant factors. Most importantly, protein S, a vitamin K-dependent glycoprotein, not only has a major role in the anti-coagulation pathway, but also prevents the production of inflammatory cytokines associated with the cytokine storm observed in acute lung, liver and heart injury seen in COVID-19 patients [208]. Low protein S levels, often due to pneumonia-induced vitamin K depletion, were correlated lately with higher thrombogenicity, clinical severity, and fatal outcome in COVID-19 patients, independently of age or inflammatory biomarkers [209]. Furthermore, ample evidence has shown that vitamin K exerts a potent calcification-inhibitory function by allowing carboxylation of Matrix Gla Protein [210].

A summary regarding the relationship among vitamins, CV diseases, and COVID-19 is reported in Table 2.

## 5. Conclusions

Although the human body needs very small quantities of micronutrients, their importance for human physiology is relevant [211]. The deficiency of electrolytes and vitamins is widespread in hospitalized patients, and it is strongly related to potential life-threatening conditions. Adequate nutrition and supplementation are pivotal to reduce the risks associated with micronutrients imbalance [211].

Micronutrient imbalance has been extensively observed in patients with CV disease and COVID-19, significantly worsening their prognosis. The causal relationship between COVID-19 and cardiovascular diseases is mutual. In fact, COVID-19 has been shown to affect the CV system extensively and, at the same time, patients affected by a CV disease are more prone to develop a severe infection form. Apart from the already well-known pathophysiological mechanisms promoting CV diseases in the setting of COVID-19 infection, derangements in the concentration of several important micronutrients, such as ions and vitamins, may have a non-negligible role in the pathophysiological continuum between these two entities. In particular, hypokalemia is a very common electrolytes imbalance in COVID-19, and it is associated with cardiac arrythmias [84,89]. Hyponatremia is an independent risk factor for mortality in COVID-19 patients, related to worse patient’s prognosis affecting blood volume regulation, osmotic equilibrium, and arterial pressure [38,40]. Hypomagnesemia at hospital admission has been associated with a higher risk of mortality and arrythmias in COVID-19 [63,64]. Vitamin D deficiency has been associated with thromboembolic events [199,200], endothelial dysfunction, left ventricular hypertrophy [189], and it is more frequent in patients admitted to ICU. Vitamin A deficiency is associated with COVID-19 symptoms severity and lipid metabolism imbalance [166,169], while vitamin E and C are closely related to good functioning of the immune system [177,201] and cardioprotection [181,182,204,205,206].

## Figures and Tables

**Figure 1 nutrients-14-03439-f001:**
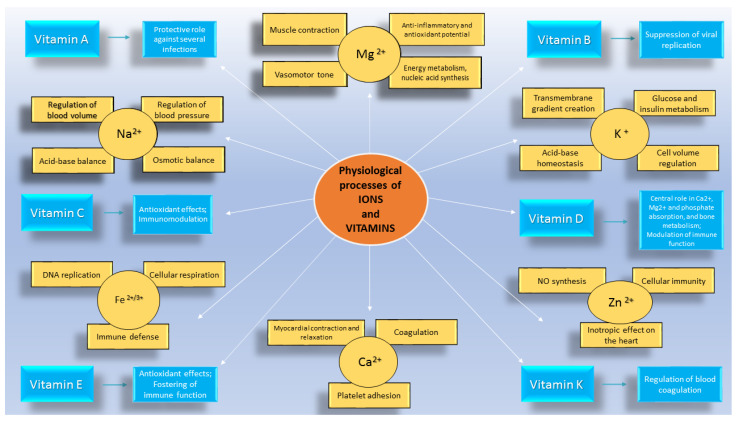
Micronutrients imbalance may contribute to cardiovascular complications observed during COVID-19. NO: nitric oxide; DNA: deoxyribonucleic acid; Mg^2+^: magnesium; K^+^: potassium; Zn^2+^: zinc; Ca^2+^: calcium; Fe^2+/3+^: iron; Na^2+^: sodium.

**Table 1 nutrients-14-03439-t001:** A summary of the most important ions imbalance during COVID-19 and their effects on CV system.

Ion	Relationship among CV Diseases and Ions	Relationship among COVID-19 and Ions	Reference
**Sodium (Na+)**	Sodium imbalance is the most common electrolyte disorder Sodium imbalance affects blood volume, blood pressure, osmotic and acid-base equilibrium regulation	Dysnatremia is an independent risk factor for mortality in hospitalized patients, including COVID-19 patients Hyponatremia is associated with increased risk of encephalopathy and mechanical ventilation necessity	[38,40]
**Magnesium (Mg++)**	Magnesium deficiency is associated with increased incidence of arrhythmias, arterial hypertension and atherosclerosis	Hypomagnesaemia has been associated with more severe COVID-19 cases	[63,64]
**Potassium (K+)**	Alterations in potassium levels predispose to cardiac arrhythmias Potassium imbalance leads to derangements in vascular tone and systemic blood pressure regulation	High incidence of potassium imbalance has been found in COVID-19 patients Hypokalemia is the most common disorder, and it is associated with worse prognosis Hyperkalemia occurs less frequently than hypokalemia	[84,89]
**Calcium (Ca++)**	Hypocalcemia is associated with derangements in coagulation and platelet adhesion Hypocalcemia affects myocardial contractility and relaxation	COVID-19 patients have lower calcium levels compared to patients without COVID-19	[106,114,115]
**Iron (Fe++/Fe+++)**	Iron deficiency and anemia are frequent comorbidities in CV diseases In HF anemia and iron deficiency represent an independent condition predisposing to CV and overall mortality, rehospitalization and quality of life and symptoms worsening	Significantly lower levels of hemoglobin have been found in COVID-19 patients admitted to ICU compared to those who were not admitted to ICU Iron deficiency is crucial in depressing immune response	[128,129,130,131,133]
**Zinc (Zn++)**	Zinc deficiency causes oxidative stress, laying the groundwork for atherosclerosis. Zinc deficiency can reduce cardiac inotropism.	Zinc supplementation can promote the restoration of ACE2 expression in COVID-19 patients.	[158,159]

COVID-19: coronavirus disease 2019; CV: cardiovascular; ICU: intensive care unit; HF: heart failure; ACE2: angiotensin-converting enzyme 2.

**Table 2 nutrients-14-03439-t002:** A summary of the most important vitamins imbalance during COVID-19 and their effects on CV system.

Vitamin	Relationship among CV Diseases and Vitamins	Relationship among COVID-19 and Vitamins	Reference
**Vitamin A**	Improvement of dyslipidemia. Reduction of overall mortality in CV diseases	Low vitamin A serum levels are associated with COVID-19 symptoms severity	[166,169]
**Vitamin B**	Reduction in homocysteine responsible for increased risk of coronary artery disease and stroke (B6, B9, B12). Protection against metabolic syndrome (B1). Antioxidant effects and improvement of endothelial function (B9).	Suppression of viral replication in host cells (B12) Vitamin B12 therapy could reduce severe damages induced by COVID-19 and related symptoms.	[171,172,176,178]
**Vitamin C**	Antioxidant effect, contrasting ROS and inflammation	Immunomodulant activity on T and B cells Antithrombotic activity through platelet expression of CD40 ligand.	[181,182,186]
**Vitamin D**	Anti-fibrotic and anti-hypertrophic role. Regulation of RAAS.	Vitamin D deficiency is associated with an increase of thrombotic episodes.	[191,192,193,194,195,196,199]
**Vitamin E**	Antioxidant effect. Inhibition of LDL oxidation. Reduction of cardiomyocytes apoptotic activity.	Suppression of PGE2 synthesis, an important T cell-suppressing mediator.	[199,201,203,204]
**Vitamin K**	Inhibition of cardiovascular calcification Anti-coagulation role	Activation of Protein S and inhibition of cytokine storm.	[208,209,210]

COVID-19: coronavirus disease 2019; CV: cardiovascular; ROS; reactive oxygen species; RAAS: renin-angiotensin-aldosterone system; PGE2: prostaglandin E2; LDL: low density lipoprotein.

## Data Availability

Not applicable.

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
