# Peer review of "The Mutual Relationship among Cardiovascular Diseases and COVID-19: Focus on Micronutrients Imbalance"

_nutrients, 2022, doi:10.3390/nu14163439_

Round 1

Reviewer 1 Report

Dear Authors,

This review aimed to analyze the role of micronutrient imbalance in the interaction between COVID-19 and cardiovascular (CV) disease. It is well known that micronutrients are involved in several physiological pathways, and their imbalance is associated with many CV disorders, including body fluid imbalances, excitability, sensitivity, cell damage, etc. Also, micronutrient imbalances associated with COVID-19 have been found to increase the risk of cardiovascular events and mortality. This review highlights the important issue of preventing excess mortality in COVID-19 due to micronutrient imbalances. Currently, this problem is extensively investigated, and it is covered in plenty of similar publications.

The authors presented the role and of ions (Na+, K+, Ca+, Fe2/3+, and Zn2+) and vitamins ( A, B, C, D, E) in CV diseases and COVID-19. However, some important aspects of their impact were missed.

1. I would recommend presenting the distribution of ions in the body (extracellular/intracellular). Figure 1 looks rather symbolic and uninformative. I propose to develop a circuit diagram that shows the participation of ions and vitamins in specific physiological processes. What was the reason for choosing only the listed vitamins and not others?

2. It should be noted whether micronutrient imbalances are characteristic of severe COVID-19; micronutrient imbalance exists prior to or appears during COVID-19. What micronutrient imbalance affects the outcome of COVID-19?

2. A typical symptom of COVID-19, such as fever, was completely absent from the review. In my opinion, the effect of fever on CV function and micronutrient imbalance in COVID should be considered and discussed in more detail. Fever is known to affect the distribution of fluids in the body as well as the function of the cardiovascular system. The authors noted the hypothalamic vasopressin secretion caused by Interleukin 1β (IL-1β) and 6 (IL-6). This might be associated with antipyretic mechanism.

3. Text beginning with “Furthermore, iron supplementation may help patients with HF [3,131,135]” (p. 10) should be edited because it is too long.

4. Paragraph 4.2 should be edited. Only the role of B12 has been discussed, while other B vitamins have not been mentioned.

5. The conclusion should be edited regarding the specific role of micronutrients in COVID-19 and CV disorders.

6. The abstract should be more meaningful and reflect the results of the analysis of the role of micronutrients in the formation of COVID-19 outcomes in patients with CV disease.

My overall comment: The manuscript in its present form is not ready for publishing. The text needs to be revised and additional analysis of the results is required.

Author Response

The authors presented the role and of ions (Na+, K+, Ca+, Fe2/3+, and Zn2+) and vitamins (A, B, C, D, E) in CV diseases and COVID-19. However, some important aspects of their impact were missed.

  • I would recommend presenting the distribution of ions in the body (extracellular/intracellular). Figure 1 looks rather symbolic and uninformative. I propose to develop a circuit diagram that shows the participation of ions and vitamins in specific physiological processes. What was the reason for choosing only the listed vitamins and not others
  • We thank the reviewer for the comment. We added a brief discussion regarding the physiological role of ions in the text. We modified the Figure 1 providing a comprehensive description of physiological role of ions and vitamins. Regarding the question provided, we had considered the listed items of greatest current interest as more used in clinical practice according to our experience, but thanks to suggestions provided we expanded the list by adding the whole group of Vitamin B and with Vitamin K.

  • It should be noted whether micronutrient imbalances are characteristic of severe COVID-19; micronutrient imbalance exists prior to or appears during COVID-19. What micronutrient imbalance affects the outcome of COVID-19?
  • We thank the reviewer for the comment. We simply reported evidence from literature. Main evidences have been demonstrated that electrolyte abnormalities occurs during hospitalization and, therefore, do not exist before so noticeably. The presence of micronutrients disorders impact on prognosis of patients with COVID-19 worsening CV outcomes. The presence of fever, the state of systemic inflammation and the use of drugs such as antibiotics, diuretics etc. further contributes to abnormalities in the distribution of fluids and electrolytes. We also tried to further stress the relationship between the specific micronutrient imbalance and COVID-19 outcome in the text. 

  •  A typical symptom of COVID-19, such as fever, was completely absent from the review. In my opinion, the effect of fever on CV function and micronutrient imbalance in COVID should be considered and discussed in more detail. Fever is known to affect the distribution of fluids in the body as well as the function of the cardiovascular system. The authors noted the hypothalamic vasopressin secretion caused by Interleukin 1β (IL-1β) and 6 (IL-6). This might be associated with antipyretic mechanism.
  • We thank the reviewer for the comment. We apologise for the missing discussion regarding fever. We provide a brief discussion about fever and its impact on cardiovascular system.

  • Text beginning with “Furthermore, iron supplementation may help patients with HF [3,131,135]” (p. 10) should be edited because it is too long.
  • We thank the reviewer for the comment. We edited the text making it shorter and concise.

  • Paragraph 4.2 should be edited. Only the role of B12 has been discussed, while other B vitamins have not been mentioned.
  • We thank the reviewer for the comment. We apologise for the missing discussion regarding other vitamins of group B. We added a paragraph about all B vitamins group.

  • The conclusion should be edited regarding the specific role of micronutrients in COVID-19 and CV disorders.
  • We thank the reviewer for the comment. We edited conclusion as suggested, adding discussion regarding the role of micronutrients in COVID-19 and CV disorders.

  • The abstract should be more meaningful and reflect the results of the analysis of the role of micronutrients in the formation of COVID-19 outcomes in patients with CV disease.
  • We thank the reviewer for the comment. We edited the abstract as suggested.

Reviewer 2 Report

Paolo Severino and colleagues discussed the effect of micronutrients imbalance on the mutual relationship between COVID-19 and cardiovascular diseases. They summarized the effect of ions (sodium, magnesium, potassium, calcium, iron and zinc) as well as vitamins (vitamin A, B, C, D, E) on both COVID-19 and cardiovascular disease. The topic is well-chosen, and the manuscript is well organized. I have the following comments and suggestions:

1.     Most of the content in figure 2, 3 and 4 can be found in table 1. Please only keep either table 1 or figure 2-4.

2.     The content of figure 5 can also be found in table 2, so it is redundant.

3.     The effect COVID-19 and cardiovascular diseases is mutual, please indicate it in figure 1.

4.     Please add the prevalence of cardiovascular diseases in COVID-19 patients. 

5.     Please include “vascular thrombosis” as one of the hypotheses of the cardiovascular effect of COVID-19 in page 2, section 2.

Author Response

Paolo Severino and colleagues discussed the effect of micronutrients imbalance on the mutual relationship between COVID-19 and cardiovascular diseases. They summarized the effect of ions (sodium, magnesium, potassium, calcium, iron and zinc) as well as vitamins (vitamin A, B, C, D, E) on both COVID-19 and cardiovascular disease. The topic is well-chosen, and the manuscript is well organized. I have the following comments and suggestions:

  • Most of the content in figure 2, 3 and 4 can be found in table 1. Please only keep either table 1 or figure 2-4.
  • We thank the reviewer for the comment. We erased Figure 2-4 and kept only Table 1.

  • The content of figure 5 can also be found in table 2, so it is redundant.
  • We thank the reviewer for the comment. We erased Figure 5.

  • The effect of COVID-19 and cardiovascular diseases is mutual, please indicate it in figure 1.
  • We thank the reviewer for the comment. We modified Figure 1 also according to other reviewer comment

  • Please add the prevalence of cardiovascular diseases in COVID-19 patients.
  • We thank the reviewer for the comment. We added this information in the section 2

  • Please include “vascular thrombosis” as one of the hypotheses of the cardiovascular effect of COVID-19 in page 2, section 2
  • We thank the reviewer for the comment. We apolgise for the lack of this important mechanism involved in cardiovascular complications occurring during COVID-19. We included this information in the text.